

# Comparative electromyography analysis of subphase gait disorder in chronic stroke survivors

Nusreena Hohsoh[1], Thanita Sanghan[1], Desmond Y.R. Chong[2], Goran Stojanovic[3] and Surapong Chatpun[1]

[1] Department of Biomedical Sciences and Biomedical Engineering, Faculty of Medicine, Prince of Songkla University, Hatyai, Songkhla, Thailand
[2] Engineering Cluster, Singapore Institute of Technology, Singapore, Singapore
[3] Department of Electronics, Faculty of Technical Sciences, University of Novi Sad, Novi Sad, Serbia

Corresponding author
Surapong Chatpun,
surapong.c@psu.ac.th

## ABSTRACT

Abnormal lower limb muscle activity is the most common cause of the alterative pattern of gait in stroke survivors, resulting from spastic and paralytic muscles around the hip, knee, and ankle joints. However, the activity of the major lower limb muscles that control the legs to facilitate walking in stroke patients have not been clearly understood in each subphase of the gait. This study differentiated the characteristics of surface electromyography (sEMG) signals of lower limb muscles during four subphases of gait cycle between stroke patients and healthy subjects. Sixteen chronic stroke patients and sixteen healthy subjects were recruited. All participants completed three walking trials with a self-selected walking speed. The sEMG signals were recorded on the gluteus medius, rectus femoris, long head of biceps femoris, medial gastrocnemius, tibialis anterior, and peroneus longus muscles. The characteristics of sEMG signals were processed and analyzed in the time and frequency features, considering the first double support, single support, second double support, and swing phases of the gait cycle. The stroke patients had altered sEMG characteristics on both paretic and non-paretic sides compared to healthy subjects across the sub-phases of gait cycle for all six muscles. All time domain features of sEMG signal showed that the medial gastrocnemius muscle has the most significant impaired activity ($p < 0.05$) and affected gait disturbance during all four subphases of the gait cycle. The findings demonstrated that the medial gastrocnemius muscle had impaired activity and was most affected during all four sub-phases of the gait cycle. This indicates that sEMG of medial gastrocnemius muscle can be used to measure the improvement of gait rehabilitation.

# INTRODUCTION

Stroke is the second leading cause of death, as well as the leading cause of long-term disability worldwide (*Katan & Luft, 2018*). Abnormal muscle activity is one of the most common impairments that leads to impaired function in the lower limb along with an abnormal gait pattern after stroke (*Li, Francisco & Zhou, 2018*). Generally, spasticity and weakness of the gastrocnemius and soleus which contribute to the ankle plantar flexion

in the mid-stance cause the abnormal gait pattern (*Li, 2020*). The hyperactivity of the quadriceps femoris and inadequate biceps femoris activation result in a stiff knee gait, which leads to reduced knee flexion in the swing phase during gait (*Wang et al., 2017*; *Akbas et al., 2020*). The most common technique for assessing the muscle activity of the lower limbs is surface electromyography (sEMG), which could be used to examine the dynamic muscle contraction that occurs during normal and abnormal gait (*Papagiannis et al., 2019*).

Abnormal muscle activities at the lower limb muscles in stroke patients have been reported using sEMG feature analysis. The root mean square (RMS) of sEMG amplitude in rectus femoris, long head of biceps femoris, and lateral gastrocnemius muscles on the paretic side of the stroke patients was found to associate with the range of motion of knee flexion (*Wang et al., 2017*). For chronic stroke patients with hemiparesis, the level of biceps femoris activity during the swing phase affected gait velocity (*Fujita, Hori & Kobayashi, 2018*). The chronic stroke with a lower number of muscle modules had a decreased gait function and a greater kinematic asymmetry (*Shin et al., 2021*). Furthermore, the activation of the motor unit responsible for the peroneus muscle lacked when the tibialis anterior contracted, leading to foot inversion in stroke patients (*Liu et al., 2020*). As noted in previous studies, the characteristics of EMG in most lower limb muscles have not been studied in detail, such as the muscles that contributed to gait during double and single supports of the stance phase and during the swing phase of the post-stroke patients.

The purpose of this study was to identify and differentiate the characteristics of sEMG signals of lower limb muscles during four subphases of gait cycle between chronic stroke patients and healthy subjects. The effects of stroke were assessed in time and frequency domains of sEMG signals from six major lower limb muscles which include gluteus medius (GM), rectus femoris (RF), long head of bicep femoris muscle (BF), medial gastrocnemius (MG), tibialis anterior (TA), and peroneus longus muscle (PL). As the muscle weakness and abnormal walking on the paretic side, we hypothesized that features of sEMG signals of these six major lower limb muscles on a paretic side exhibit lower values than that on a non-paretic side and significantly differ as compared to a non-paretic side in double support and swing subphases of gait cycle.

## MATERIALS & METHODS

### Participants

This study was conducted at the Southern Medical Rehabilitation Center of the Songklanagarind Hospital, Songkhla, Thailand. We recruited 16 chronic stroke patients and 16 healthy subjects with age above 45 years old. All stroke patients were able to walk at least 30 m without the use of walking aids. In addition, all of them had the ability to follow the step command, had no previous history of musculoskeletal problems and neurological conditions, and did not receive any lower limb arthroplasty. All healthy subjects with no history of musculoskeletal problems, neurological conditions, and lower limb arthroplasty were enrolled. This work was approved by the human research ethic committee of the Faculty of Medicine, Prince of Songkla University (REC.64-386-25-2). All participants provided signed informed consent.
## Data collection and experimental protocol

All participants were asked to perform a 10-meter walk test and complete the Montreal Cognitive Assessment (MoCA). Twelve channels of the electromyographic system (Zerowire, Aurion, Italy) were used to record EMG signals with a sampling rate of 1,000 Hz. Three force plates (AccuGait Force Plate; AMTI, Watertown, MA, USA) were used to record the ground reaction force with a sampling rate of 1,000 Hz. The Pedar-X in-shoe pressure measurement system (Novel, Munich, Germany) was used to detect the gait cycle, *i.e.,* stance phase (ST), first double support (DS1), single support (SS), second double support (DS2), and swing phase (SW) with a sampling rate of 100 Hz. The recorded data were manually synchronized between force plates and the Pedar-X system from the gait events and then mapped the gait cycles to EMG signals.

The Ag/Cl bipolar surface electrodes (Ambu® surface electrodes; Ambu Sdn. Bhd., Penang, Malaysia) with a skin contact diameter of 34 mm were placed on six muscles on both lower limbs in the specific locations of the motor point according to SENIAM protocol. The lower limb muscles of interest were gluteus medius (GM), rectus femoris (RF), long head of bicep femoris muscle (BF), medial gastrocnemius (MG), tibialis anterior (TA), and peroneus longus muscle (PL). Figure 1 shows the placement of EMG electrodes.

All participants completed three walking trials, walking back and forth along a 10-meter walkway for two rounds with a self-selected walking speed. They took a rest in a sitting position for 5 min between the trials along with assessing their vital signs.

## Data processing and analysis

The raw sEMG signals were filtered by fourth order band pass Butterworth filter 30–450 Hz (*Hwang, Oh & Jeon, 2018*; *Wei et al., 2021*) and the power line noise was eliminated using a notch filter with a frequency of 50 Hz. sEMG data from the left and right sides of the healthy subjects were combined for analysis. For the stroke patients, the sEMG signals on the paretic and non-paretic sides were analyzed separately. The lower limb muscles activities were evaluated in four sub-phases of the gait cycle: DS1, SS, DS2, SW phases. In this study, five consecutive gait cycles in each subphase of gait cycle were analyzed in each participant (*Chen et al., 2020*). The muscle activity in each gait phase was normalized with the mean of absolute sEMG across five consecutive gait cycles using Eq. (1) (*Phinyomark, Phukpattaranont & Limsakul, 2012*).

$$\text{Normalized sEMG} = \frac{\text{filtered sEMG in each gait phase}}{\text{Mean}\left|\text{filtered sEMG of 5 gait cycles}\right|}. \tag{1}$$

The sEMG signals were examined using both time-domain and frequency-domain analyses. The parameters used in the time domain analysis were the root mean square (RMS), mean absolute value (MAV), log detector (LOG), and waveform length (WL). The mean frequency (MNF) and median frequency (MDF) (*Swank et al., 2020*; *Wang et al., 2021*) were used in the frequency domain analysis.

 

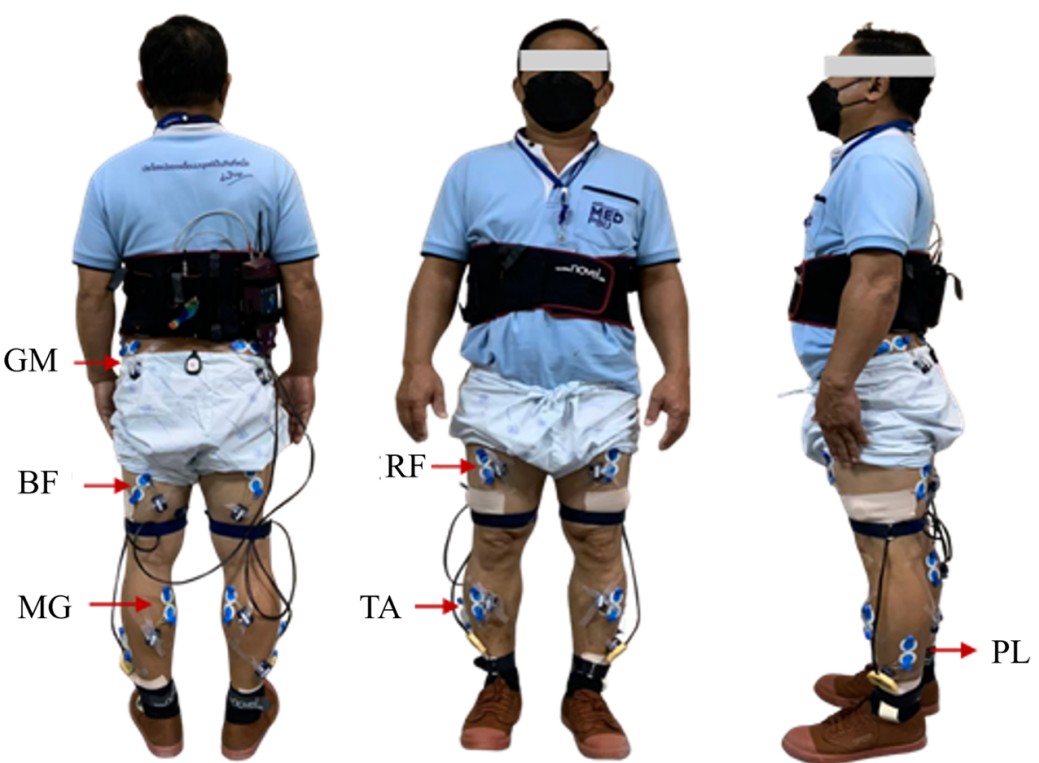

**Figure 1** Electrode placement for EMG recording included the gluteus medius (GM), rectus femoris (RF), biceps femoris (BF), medial gastrocnemius (MG), tibialis anterior (TA), and peroneus longus (PL) muscles.

The RMS was used to estimate the non-fatiguing and constant force of muscle contraction.

$$\text{RMS} = \sqrt{\frac{1}{N}\sum_{i=1}^{N} x_i^2} \tag{2}$$

where $x_i$ is the sEMG signal amplitude at each time point ($i$), and $N$ is the number of sampling points.

The MAV was used to evaluate the onset and offset timing of muscle activation.

$$\text{MAV} = \frac{1}{N}\sum_{i=1}^{N} |x_i|. \tag{3}$$

The LOG was used to determine the muscle contraction force.

$$\text{LOG} = e^{\frac{1}{N}\sum_{i=1}^{N}\log(|x_i|)}. \tag{4}$$

The WL was used to describe the sEMG waveform's cumulative length over the time segment.

$$\text{WL} = \sum_{i=1}^{N-1} |x_{i+1} - x_i|. \tag{5}$$

The MNF was used to determine the motor unit recruitment of the muscles.

$$\text{MNF} = \frac{\sum_{j=1}^{M} f_j P_j}{\sum_{j=1}^{M} P_j} \tag{6}$$

where $f_j$ represents the frequency of the spectrum at frequency bin $j$, $P_j$ represents the EMG power spectrum at frequency bin $j$, and $M$ represents number of the frequency bin.

The MDF was used to determine the motor unit (MU) recruitment of the muscles.

$$\sum_{j=1}^{MDF} P_j = \sum_{j=MDF}^{M} P_j = \frac{1}{2} \sum_{j=1}^{M} P_j. \tag{7}$$

## Statistical analysis

The parameters of muscle activity were compared using (Prism 9.0; GraphPad Software, San Diego, CA, USA). A Mann–Whitney u test was employed to compare differences between healthy subjects and stroke patients. A Wilcoxon matched pairs signed rank test was performed in a case involving paretic and non-paretic sides. A statistical significance was considered if $p < 0.05$.

# RESULTS

## Characteristics of the participants

Sixteen stroke patients and sixteen healthy subjects were recruited for this study. The characteristics of the participants' data are presented in Table 1. Stroke patients took significantly longer time for 10-meter walk test than healthy subjects ($p < 0.0001$).

## Electromyography signals

The differences in amplitude, duration, and pattern of filtered sEMG signals of six lower limb muscles were observed between the healthy subject and the stroke patient (non-paretic and the paretic sides) in DS1, SS, DS2, and SW phases as shown in Fig. 2.

## Characteristics of muscle activity

### Gluteus medius (GM)

Figure 3A shows the features of GM activity. During DS1 phase, the values of RMS, MAV, LOG, and WL on paretic and non-paretic sides are significantly lower compared to healthy subjects ($p < 0.05$). In the SS phase, the value of WL in the paretic side was significantly higher than the non-paretic side and healthy subjects ($p < 0.05$) whereas the value of MNF on the paretic side was significantly lower than the non-paretic and healthy subjects ($p < 0.05$). In the DS2 phase, the paretic side showed significantly higher RMS than the non-paretic ($p < 0.05$). Furthermore, the RMS, MAV, LOG and WL of GM on a non-paretic side were lowest among the groups in the SW phase.

### Rectus femoris (RF)

The paretic side of stroke patients exhibited significantly lower RMS, MAV, LOG, and WL values in the RF muscle compared to healthy subjects ($p < 0.05$) during the DS1 phase (Fig. 3B). Furthermore, the paretic side demonstrated significantly lower values of MNF
**Table 1  Demographics and test characteristics of the participants.**

| Variables | Stroke ($N = 16$) | Healthy ($N = 16$) |
|---|---|---|
| Age (Years) | $61.06 \pm 7.1$ | $60.75 \pm 7.14$ |
| Gender (Males/Females) | 11/5 | 9/7 |
| Body mass index (Kg/m$^2$ ) | $24.55 \pm 3.58$ | $23.39 \pm 3.05$ |
| 10-meter walk test (Second) | $27.48 \pm 13.15$[****] | $9.13 \pm 1.78$ |
| MoCA score (Point) | $21.36 \pm 9.55$ | $25.75 \pm 2.62$ |
| Time post-stroke (Months) | $47.12 \pm 43.43$ | – |
| Stroke type (Ischemic/Hemorrhagic) | 11/5 | – |
| Paretic side (Left/Right) | 8/8 | – |
| Manual muscle test (non-paretic/paretic) | | |
| Hip flexor | $9.67 \pm 0.78/7.00 \pm 1.48$ | – |
| Hip extensor | $9.67 \pm 0.78/7.25 \pm 1.36$ | – |
| Knee flexor | $9.83 \pm 0.58/5.58 \pm 2.57$ | – |
| Knee extensor | $9.83 \pm 0.58/6.08 \pm 2.11$ | – |
| Ankle dorsiflexor | $9.67 \pm 0.78/3.25 \pm 2.42$ | – |
| Ankle plantarflexor | $9.67 \pm 0.78/3.50 \pm 2.75$ | – |

Notes.

Values are presented as mean $\pm$ standard deviation
[****] Significant difference between stroke patients and healthy subjects ($p < 0.0001$).
The Manual muscle test was graded on a 10-point scale (1–10).

and MDF compared to the healthy subjects ($p < 0.05$). In the SS phase, the paretic side of stroke patients had RMS, MAV, LOG, and WL significantly higher than the healthy subjects ($p < 0.05$). The values of MAV and LOG were significantly higher on the non-paretic side compared to the healthy subjects ($p < 0.05$). In contrast, the MNF and MDF of RF were significantly lower on the paretic and the non-paretic sides compared to the healthy subjects ($p < 0.05$).

### Long head of biceps femoris (BF)

The results shown in Fig. 4A indicate that the RMS, MAV, LOG, and WL of BF were significantly lower on the paretic side compared to the non-paretic side and the healthy subjects in the DS1 phase ($p < 0.05$). In contrast, in the SS phase, the paretic side exhibited significantly higher values of RMS, MAV, LOG, and WL than healthy subjects ($p < 0.05$). In the DS2, the RMS, MAV, LOG, and WL of BF were significantly higher on the paretic side and the non-paretic side compared to the healthy subjects ($p < 0.05$). In the SW phase, the paretic side had significantly lower RMS, MAV, LOG and WL than the healthy subjects ($p < 0.05$). In contrast, the MNF and MDF of BF on the paretic side were significantly higher than the non-paretic side and the healthy subjects ($p < 0.05$).

### Medial gastrocnemius (MG)

Figure 4B illustrates features of sEMG of MG in four different phases of a gait cycle. In the DS1 phase, the RMS, MAV, LOG, and WL of MG of non-paretic side were significantly higher than that of MG of healthy subjects ($p < 0.05$). In the SS phase, the paretic side had significantly lower than the healthy subjects in the RMS, MAV, LOG, WL of MG and the non-paretic side in the RMS of MG ($p < 0.05$). In the DS2 phase, the paretic side

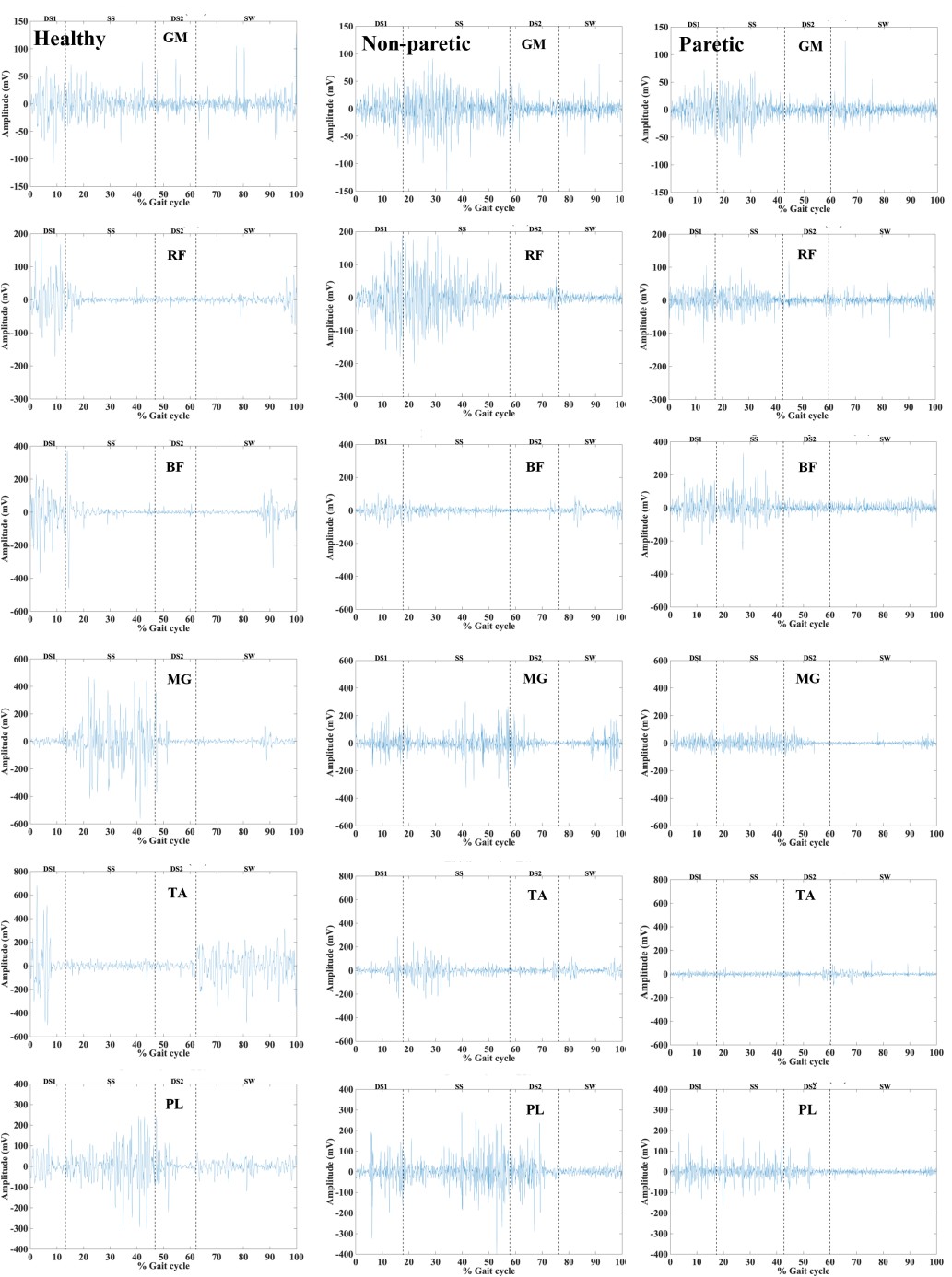

**Figure 2** Selected filtered EMG signals of six lower limb muscles of a healthy subject (Left panel) and stroke patients on a non-paretic side (middle panel), paretic side (right panel) during the first double support (DS1), single support (SS), second double support (DS2), and swing(SW) phases.

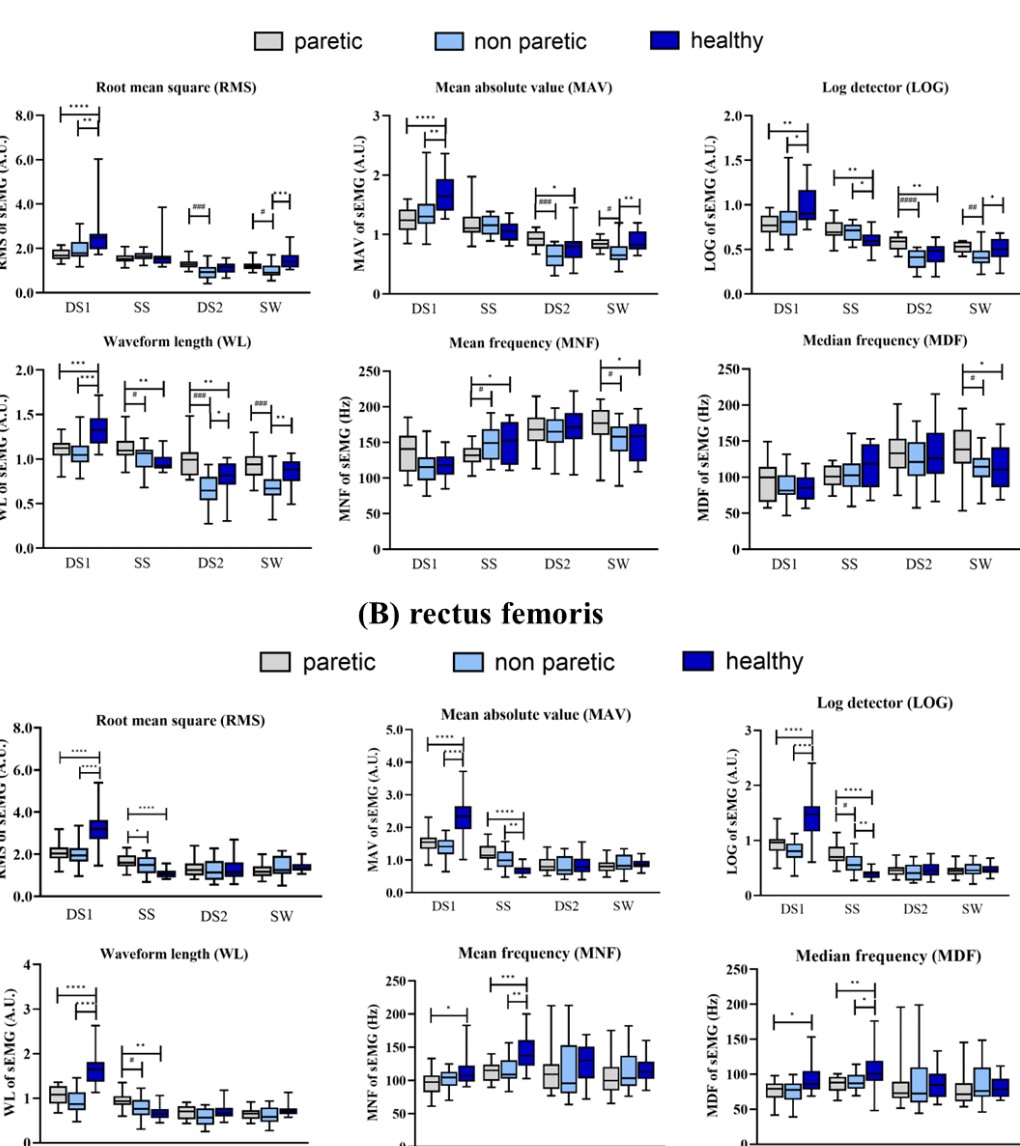

**Figure 3** The gluteus medius (A) and rectus femoris (B) muscles activity during DS1: 1st double support, SS, single support; DS2, 2nd double support and SW, swing phases paretic *vs* non-paretic: # *p* < 0.05; ## *p* < 0.01; ### *p* < 0.001; #### *p* < 0.0001 and between group: \**p* < 0.05; \*\**p* < 0.01; \*\*\**p* < 0.001; \*\*\*\**p* < 0.0001.

had significantly higher than the healthy subjects in the RMS, MAV, LOG, WL of MG ($p < 0.05$) and the non-paretic side in the MNF and MDF ($p < 0.05$). Lastly, in the SW phase, the paretic side exhibited significantly higher values of RMS, MAV, LOG, and WL in the MG muscle compared to both the non-paretic side and healthy subjects ($p < 0.05$).

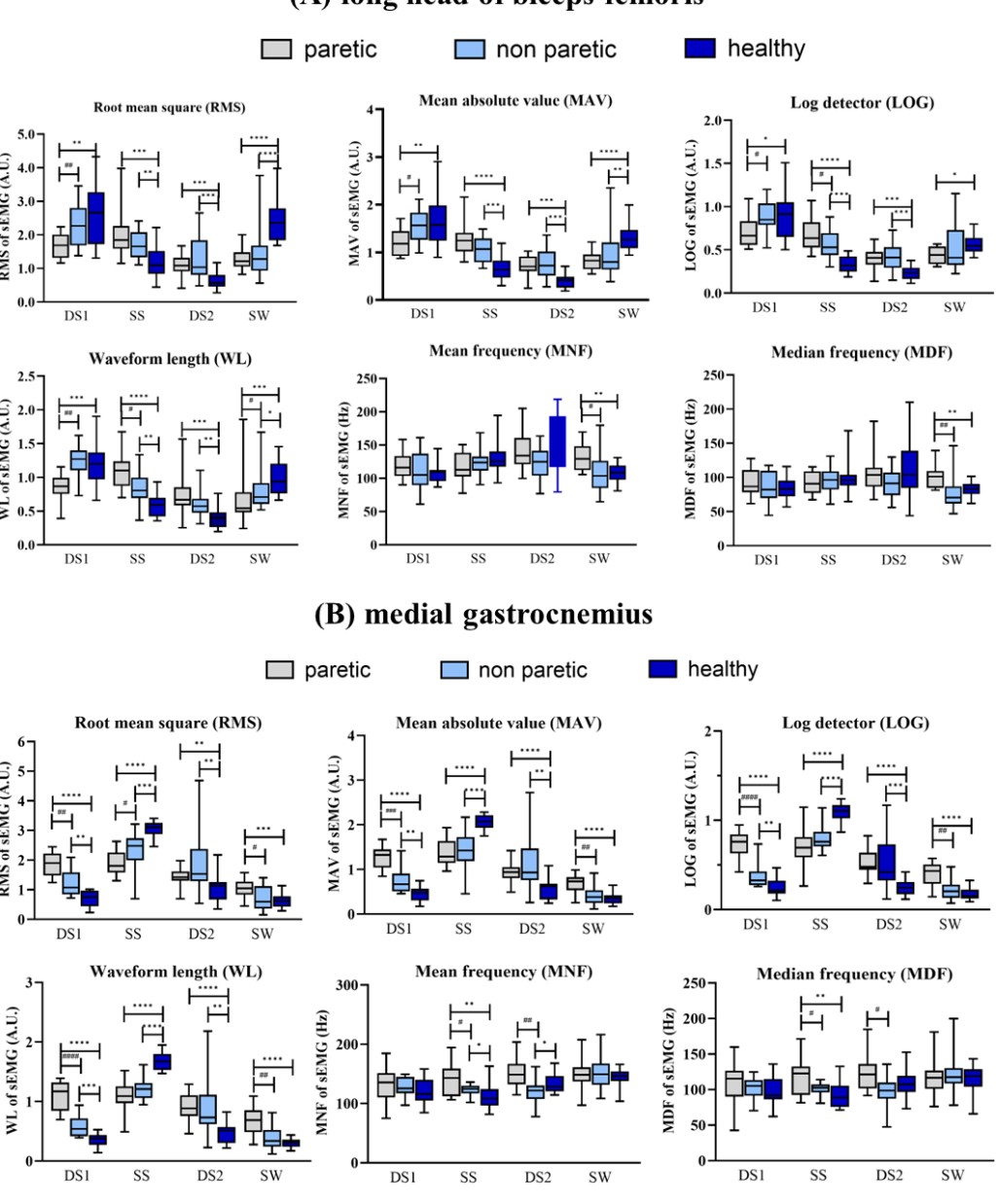

**Figure 4  The long head of biceps femoris (A) and medial gastrocnemius (B) muscles activity during DS1, 1st double support; SS, single support; DS2, 2nd double support and SW, swing phases paretic *vs* non-paretic: # $p < 0.05$; ## $p < 0.01$; ### $p < 0.001$; #### $p < 0.0001$ and between group: *$p < 0.05$; **$p < 0.01$; ***$p < 0.001$; ****$p < 0.0001$.**

### Tibialis anterior (TA)

In the DS1 phase, the RMS, MAV, LOG, and WL of TA on the paretic side was significantly lower compared to both the non-paretic side and healthy subjects ($p < 0.05$) as presented in Fig. 5A. In the SS phase, the RMS, MAV, LOG, and WL of TA on the non-paretic side were significantly higher than that on the healthy subjects ($p < 0.05$) whereas the MNF

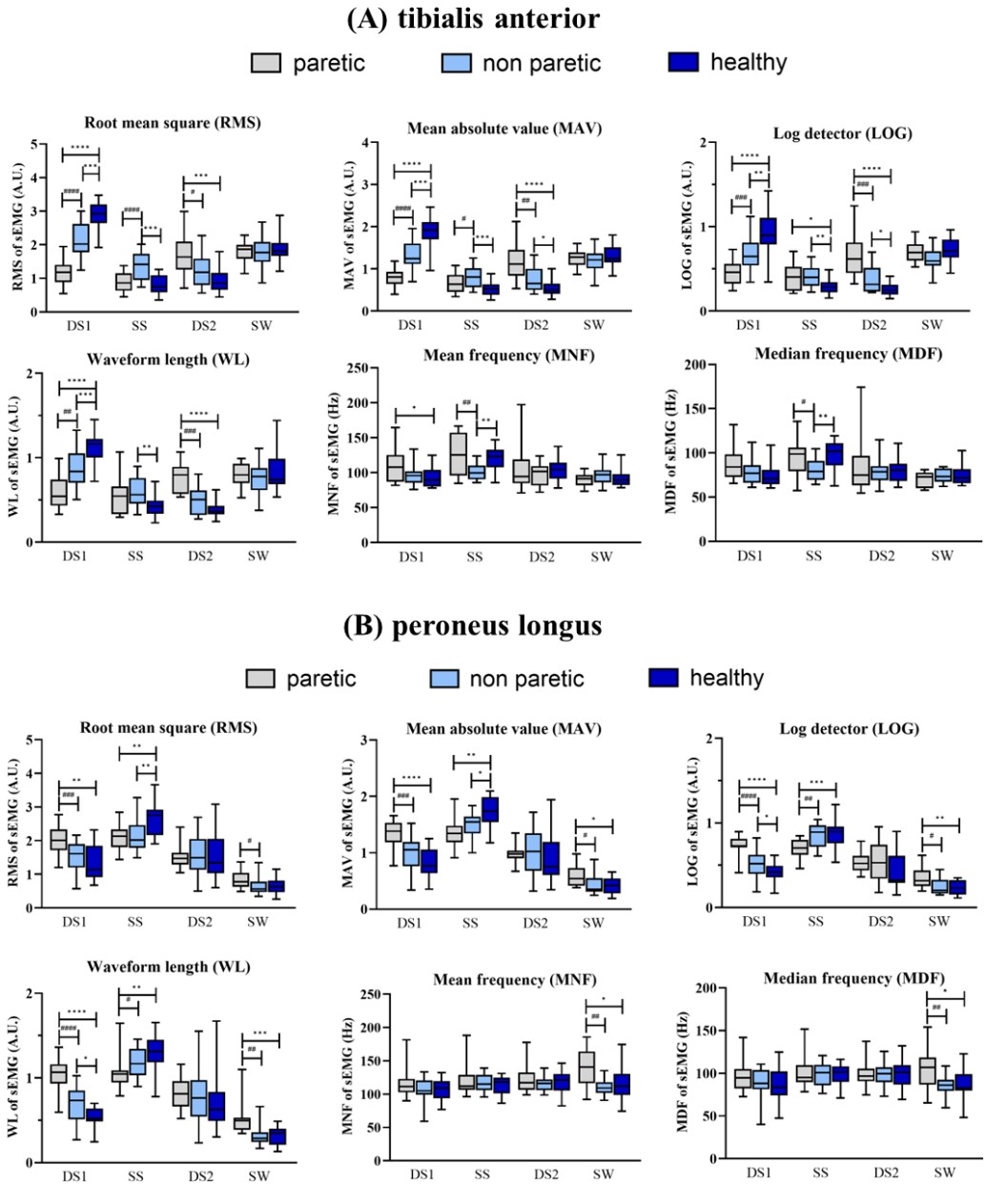

**Figure 5** The tibialis anterior (A) and peroneus longus (B) muscles activity during DS1, 1st double support; SS, single support; DS2, 2nd double support and SW, swing phases paretic *vs* non-paretic: # $p < 0.05$; ## $p < 0.01$; ### $p < 0.001$; #### $p < 0.0001$ and between group: *$p < 0.05$; **$p < 0.01$; ***$p < 0.001$; ****$p < 0.0001$.

and MDF of TA on the non-paretic side were significantly lower than that on the healthy subjects ($p < 0.05$). In the DS2 phase, the paretic side had significantly higher RMS, MAV, LOG, and WL compared to the non-paretic side and the healthy subjects ($p < 0.05$). In the SW phase, no significant difference of features was found.

### Peroneus longus (PL)

Figure 5B presents the sEMG characteristics of the PL muscle during four different phases of a gait cycle. In the DS1 phase, the paretic side had significantly higher RMS, MAV, LOG, and WL than the non-paretic side and the healthy subjects ($p < 0.05$). In the SS phase, the RMS, MAV, LOG, and WL of PL on the paretic side was significantly lower than that on the healthy ($p < 0.05$). In the DS2 phase, no significant difference of features was found. The MAV, LOG, WL, MNF, and MDF of PL on the paretic side were significantly higher than that on the healthy subjects during the SW phase ($p < 0.05$). Furthermore, the paretic side had significantly higher RMS, MAV, LOG, WL, MNF, and MDF of PL compared to the non-paretic side ($p < 0.05$).

## DISCUSSION

This study investigated the variations in sEMG signals of six lower limb muscles during different stages of a gait cycle between stroke patients and healthy subjects. The results were analyzed in both time and frequency domains. We found that both the amplitude and activated time of the sEMG signals of the 6 muscles on paretic side during DS1 and SS were different when compared to healthy subjects (especially BF and MG). Our results indicated the stroke patients had a lower muscle activity of the GM, RF, BF, and TA but higher muscle activity of the MG and PL on both paretic and non-paretic sides compared to the healthy subjects during the first double support. Additionally, the patients had deficient RF, BF, and TA contraction on both the paretic and non-paretic sides (*Chen, 2014*). During the extended walking test, stroke patients had the EMG amplitudes of TA and RF on the paretic and non-paretic sides significantly higher compared to healthy subjects (*Fujita, Kobayashi & Hitosugi, 2021*). Decreasing amplitude and increasing frequency in TA, along with increasing amplitude in MG and PL on the paretic side, could be a compensatory mechanism for "insufficient" ankle muscle co-contraction to improve balance (*Souissi et al., 2018*).

In the single support phase, the MG and PL were lower in muscle contraction but higher in the RF and BF muscles in both paretic and non-paretic sides in comparison to the healthy subjects. Lower ankle co-contraction may occur if weight-bearing on the paretic side is reduced. The weakness of muscles at the ankle joint caused by a significant loss of strength may be compensated by increasing the motor input of the knee muscles to ensure stability (*Souissi et al., 2018*). A decrease in EMG frequency despite an increase in EMG amplitude appears when motor units are synchronized. Thus, synchronizing the motor units in the muscles on the paretic side could enhance or maintain walking performance (*Fujita, Kobayashi & Hitosugi, 2021*). We observed an increase in EMG frequency with a decrease in EMG amplitude of the MG on both paretic and non-paretic sides compared to the healthy subjects during the single support. This indicated that a spasticity of MG on the paretic side might occur (*Tan et al., 2020*). Our results showed that the paretic side had higher MNF and MDF in GM, BF and PL than non-paretic side during the SW phase. This particularly relates to an increased motor unit recruitment of the lower limb's muscles.

It was observed in the study that, in a second double support, the MG and TA had a high muscle contraction level in the paretic and non-paretic sides compared to normal

lower limb. A previous study reported that the ankle muscles in stroke patients were highly activated because of a high co-contraction of the ankle muscles to provide the forward propulsion of the body (*Souissi et al., 2018*). The MG contributed to knee flexion and extension in a second double support depending on the joint kinematic position of an individual (*Brough, Kautz & Neptune, 2022*).

A reduction in knee flexion during the swing phase is one of stroke patients' most prevalent gait abnormalities. One possible contributing factor is the weakness of BF which is a knee flexor muscle, and stiff knee gait is a typical pattern of this disorder (*Balaban & Tok, 2014*). Our study demonstrated that, in the SW phase, there was a decrease in RMS, MAV and WL of BF on both paretic and non-paretic sides. Wang et al. reported that both paretic and non-paretic limbs had an impaired knee flexion (*Wang et al., 2017*). It has been reported that pedaling can increase knee flexion during swing phase in hemiparetic patients with stiff knee gait (*Fujita et al., 2020*). The non-paretic side shortened the swing time to compensate for the weakened paretic limb. An increase in the antagonist of ankle plantar flexor activation which relates to MG was associated with knee extension during the swing phase, whereby an agonist tibialis anterior motor neuron was inhibited by the excessive MG activation. In addition, muscle activation of the TA muscle was increased because of adapting behavior to overcome excessive passive plantar flexor resistance (*Ghédira et al., 2021*). *Rozanski et al. (2020)* reported that stroke patients who walk with an asymmetric gait exhibited the MG burst activity on the paretic side during the swing phase. An imbalance between TA and PL causes hind-foot varus. The peroneus activation must compensate for the physiological varus position associated with tibialis anterior contraction (*Deltombe et al., 2017*). Furthermore, to compensate excess ankle plantarflexion, stroke patients had increasingly hip abduction (*Akbas et al., 2019*).

The frequency spectrum is shifted downward when the muscle becomes fatigued because the muscle fatigue reduces a maximal force or a power production to produce a muscle contraction (*Phinyomark et al. (2012)*). *Lung et al. (2021)* used the MDF feature to reflect muscle fatigue at various walking intensities. *Toro et al. (2019)* measured the muscle fatigue from the MNF feature of the RF muscle while performing isometric contraction. Our study demonstrated that MNF and MDF features could differentiate between stroke patients and healthy subjects. Therefore, the results of the MNF and MDF from current study will be useful in determining muscle fatigue for further study using machine learning to predict myoelectric biomarkers in post-stroke gait (*Hussain & Park, 2021*).

Our finding indicates using RMS feature of sEMG as a monitoring parameter for muscle activation level can be beneficial for the rehabilitation intervention of stroke patients. In early recovery after rehabilitation intervention, sEMG signals of impaired muscles are possible to be changed or varied due to compensatory muscle activity. As rehabilitation progresses, muscle activation levels and muscle recruitment timing tend to become more efficient and improved. These changes of EMG metrics are important indicators for clinicians to monitor the progress of recovery.

This study has some limitations. Firstly, the stroke patients had heterogeneous brain lesions, stroke type, and gait patterns which may affect the characteristics of the sEMG signals. Secondly, this study only enrolled chronic stroke patients who had a stroke for

more than six months. There was high variation of post-stroke time in the patient group. The current findings may not be applicable to subacute or acute stroke. Furthermore, the interaction between these muscles should be further investigated in stroke patients during walking.

## CONCLUSIONS

The stroke patients had a lower muscle activity of the GM, RF, BF, and TA but higher muscle activity of the MG and PL on both paretic and non-paretic sides compared to the healthy subjects during the first double support. It showed that the medial gastrocnemius muscle had impaired activity and was most affected during all four sub-phases of the gait cycle. Stroke patients had excessive activation of the medial gastrocnemius and peroneus longus muscles during swing phase which was found on both paretic and non-paretic sides. Among six sEMG parameters investigated, the RMS of sEMG signals was unique in all muscles and all sub-phases and could be used to differentiate between stroke patients and healthy subjects, especially the RMS of MG electromyography. This differentiation can be used as indicators for the improvement of gait rehabilitation and for further gait rehabilitation management for stroke patients.

## ACKNOWLEDGEMENTS

We would like to thank all participants for their time in this study. We also would like to thank Associate Professor Dr. Pornchai Sathitrapanya and Dr. Thanyalak Amornpojnimman for their contribution to subjects' recruitment and consultation.

### Funding

This work was supported by the Graduate School of Prince of Songkla University and the Faculty of Medicine, Prince of Songkla University (REC 64-386-25-2). This work was also supported by the European Union's HORIZON-MSCA-2021-SE-01, MSCA Staff Exchanges 2021 under grant agreement no. 101086348. The funders had no role in study design, data collection and analysis, decision to publish, or preparation of the manuscript.

### Grant Disclosures

The following grant information was disclosed by the authors:
The Graduate School of Prince of Songkla University.
The Faculty of Medicine, Prince of Songkla University: REC 64-386-25-2.
The European Union's HORIZON-MSCA-2021-SE-01, MSCA Staff Exchanges 2021: 101086348.

### Competing Interests

The authors declare there are no competing interests.

## Author Contributions

- Nusreena Hohsoh conceived and designed the experiments, performed the experiments, analyzed the data, prepared figures and/or tables, authored or reviewed drafts of the article, and approved the final draft.
- Thanita Sanghan conceived and designed the experiments, performed the experiments, prepared figures and/or tables, and approved the final draft.
- Desmond Y.R. Chong analyzed the data, authored or reviewed drafts of the article, supervision, and approved the final draft.
- Goran Stojanovic conceived and designed the experiments, authored or reviewed drafts of the article, supervision, and approved the final draft.
- Surapong Chatpun conceived and designed the experiments, analyzed the data, prepared figures and/or tables, authored or reviewed drafts of the article, supervision, and approved the final draft.

## Human Ethics

The following information was supplied relating to ethical approvals (*i.e.*, approving body and any reference numbers):

The human research ethic committee of the Faculty of Medicine, Prince of Songkla University.

## Data Availability

The raw dataset of EMG signals is available at figshare: Chatpun, Surapong (2024). EMG signals. figshare. Dataset. https://doi.org/10.6084/m9.figshare.26065414.v1.

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
