# Peer review of "Comparative electromyography analysis of subphase gait disorder in chronic stroke survivors"

_PeerJ, doi:10.7717/peerj.18473_

## Round 0.1 · original submission · Minor Revisions

Please address all comments of both expert reviewers

Reviewer 1 ·

Basic reporting

1) Can you provide a figure regarding the placement of the electrodes?

2) Concerning the references, the Authors should refer also to very recent studies investigating EMG metrics in stroke and healthy controls during gait to compare their results to the most recent achievements and to contextualize their study.

Experimental design

1) Can you provide more details about how the gait cycle was defined?

2) Were the data from the three force plates used to relate the ground reaction force to EMG metrics?

3) Did the Authors employ some scale of motor function to define the motor function of the stroke patients (e.g., gross motor function), also to define the inclusion and exclusion criteria?

Validity of the findings

1) Can you provide a more detailed interpretation of the significant findings and their clinical relevance? How might rehabilitation interventions affect the EMG metrics over the course of recovery?

Reviewer 2 ·

Basic reporting

The manuscript titled "Comparative electromyography analysis of subphase gait disorder in chronic stroke survivors" presents an in-depth examination of lower limb muscle activity in chronic stroke survivors during different phases of the gait cycle. The study employs surface electromyography (sEMG) to differentiate the characteristics of muscle activity between stroke patients and healthy subjects, focusing on the gluteus medius, rectus femoris, long head of biceps femoris, medial gastrocnemius, tibialis anterior, and peroneus longus muscles.
The manuscript is well-written, with clear and professional English used throughout. While there are minor areas that could be refined to improve readability and clarity, these do not significantly impact the overall quality of the manuscript. The authors have effectively communicated their research findings in a manner that is accessible to their intended academic audience.

The authors provide a thorough background on the topic, citing relevant literature to contextualize their study. The introduction is well-structured, clearly outlining the significance of understanding electromyography (EMG) characteristics in stroke patients and the gaps in existing research. The references are appropriate and up-to-date, demonstrating a comprehensive review of the relevant literature. However, it may benefit from including more recent studies from the past two years to ensure the discussion reflects the latest advancements in the field.

The article follows a standard scientific structure, with clearly defined sections for the introduction, methods, results, and discussion. The use of figures and tables is effective, with each figure and table being relevant to the text and aiding in the reader's understanding of the study's findings. The figures are well-labeled and described in detail, though some captions could be more explanatory to be fully self-contained. The manuscript also indicates that raw data is provided, adhering to open data policies, which is a strong point. This allows for transparency and the possibility of replication, which is crucial in scientific research (e.g. Romano, F.; Formenti, D.; Cardone, D.; Russo, E.F.; Castiglioni, P.; Merati, G.; Merla, A.; Perpetuini, D. Data-Driven Identification of Stroke through Machine Learning Applied to Complexity Metrics in Multimodal Electromyography and Kinematics. Entropy 2024, 26, 578. https://doi.org/10.3390/e26070578)
The study is self-contained, with all necessary information provided within the text to understand the research conducted and the results obtained. The authors clearly state their hypotheses and provide results that directly address these hypotheses. The discussion ties the findings back to the hypotheses and the existing literature, providing a coherent narrative that supports the study’s conclusions. The results are presented in a logical order, making it easy to follow the progression from data collection through analysis to the final interpretations.

Experimental design

Line 83 and 93-- in the materials and methods section, it is necessary to better define the sample of stroke subjects. What were the inclusion and exclusion criteria? Some non-exhaustive examples:
• how long had the subjects not been treated with Botulinum Toxin, which could affect the EMG signal, or with treatments for spasticity reduction?
• Had the patients undergone any functional orthopedic surgery?
• Were the subjects undergoing rehabilitative treatment?
It would be appreciated to define in table 1, if it’s possible, muscle strength and spasticity for the muscle groups evaluated using clinical evaluation scales
Line 101 -- Was a specific protocol for sEMG signal acquisition and electrodes placement used (e.g. SENIAM)? What is the diameter of the electrodes used for sEMG?

Validity of the findings

Line 283 --- It would be appropriate to increase the limits of the study by describing the correlation between reduction/change in walking speed that results in modifications in muscle activity patterns during walking.

---

## Round 0.2 · accepted · Accept

Thank you for addressing the Reviewers comments/concerns. I am pleased to recommend your amended manuscript for publication.